# Methodologic Recommendations to Implement Pavement Management Systems and Eco-Sustainable Solutions for Local Road Administrations

Andrea Grilli and Alex Balzi *

Department of Economy, Science and Law, University of the Republic of San Marino, Via Consiglio dei Sessanta 99, 47891 Dogana, San Marino
* Correspondence: alex.balzi@unirsm.sm; Tel.: +378-0549-888111

**Abstract:** Local Road Administrations (LRA) manage wide and fragmented road networks with constrained financial and human resources. Though LRA manage the most road networks and the relative development and maintenance have a huge impact on environment and society, frequently, LRA cannot implement new technologies and methodological advancements because innovations are generally fitted to different kinds of users. For these reasons, the adoption of a customised pavement management system (PMS) for LRA is strongly recommended to define workflows, make investments, and find long-lasting and cost-effective solutions. Considering the goal of a sustainable development, new complex aspects must be also considered in the PMS matching policies, investment strategies and engineered solutions. Eco-sustainable techniques for the maintenance of road pavements and utility trenches must be gradually introduced in PMS involving stakeholders to preserve natural resources, to reduce atmospheric pollutions and to stimulate the local economic growth. The definition of a PMS guideline for LRA considering new concepts of a circular economy is a contemporary and open challenge. This paper shows a new PMS approach for LRA, including the strategy and requirements for environmentally friendly road materials that can be progressively adopted by each LRA to lessen the negative effects of maintenance activities on the environment.

**Keywords:** pavement management system (PMS); Local Road Administrations (LRA); Geographic Information System (GIS); eco-sustainable techniques

## 1. Introduction

The fundamental purpose of road maintenance is to maintain the safety and rideability of road pavements. A systematic approach of maintenance organisation allows a cost-effective managing of a road pavement network [1].

The Pavement Management System (PMS) is intended to determine a coherent workflow and a cost-effective maintenance planning [2]. The systematic process has to consider technical, economic, social and environmental impacts. Even though the fundamentals of PMS remain the same when applied to many contexts, Local Road Administrations (LRA) require specialised tools that are significantly different from those used by highway or airfield administrations [3,4]. LRA often manage the longest road networks and a wide range of road types using constrained financial, human and technological resources. The technical procedure and management techniques must be developed in accordance with the specific resources and requirements of the LRA [5,6]. It is necessary to decide on tools and methods while considering the actual potentialities of LRA [7,8]. High-tech methods often appear complex to implement by Local Road Administrations and this hinders the PMS application that, first of all, must be considered a workflow system. Indeed, the existing knowledge mainly refers to airfield and highway administrations, whereas LRA can just count few reference studies and guidelines. Currently, there are a lack of methodological

recommendations for LRA even if they manage the most road networks, and this activity has a huge impact on society's everyday life.

In recent years, many road administrations have been implementing PMS with Geographic Information System (GIS) to streamline data collecting, to apply decision-making processes and to determine strategic solutions in accordance with annual budget constraints. The management process can be set to use GIS technologies to support surveying activities, prioritising maintenance and choosing cost-effective solutions. Customised GIS-based procedures and technical manuals can be addressed to improve and standardise the data gathering, analysis and planning of maintenance activities, taking into account short- and long-term perspectives [9,10].

After the first definition of the Sustainable Development Goals by the United Nations in 2015, the European Commission has even more stimulated a new awareness on environmental and social issues, adopting the European Green Deal in 2019. In this new and complex context, every PMS has to implement specific tasks and targets to take into account the environmental and social impact of infrastructures maintenance activities. To lessen the negative effects of the maintenance work on society and the environment, eco-sustainable techniques and materials must be promoted in PMS as an alternative to conventional materials and methods [11,12]. Although environmentally friendly construction methods have some advantages over conventional methods [13–15], the use of them in road maintenance projects appears to be hindered by scepticism among government politicians, technicians and professional engineers. This is largely due to the lack of well-established knowledge and design documentation. Best practices on environmentally friendly procedures must be shared to LRA since LRA typically lack the means and stimulations to update their standards to new practices. Developing a new PMS approach including eco-sustainable purposes and features is a new challenge and can offer great perspectives on both administrative and social issues.

The methodological approach is of paramount importance in pavement management activity and, even if resources, technologies and targets can be different from municipality to municipality, the methodological approach does not change and must be clearly defined.

This paper summarises skills and tools coming from the application of tailored PMS in several municipalities in central Italy which manage road networks ranging from 200 to 1000 km, employing approximately two technicians every 200 km [8,9,16].

The paper describes general principles and standards behind the PMS to be easily adopted by LRA for the implementation of a rational data collection, monitoring and planning activities. An up-to-date PMS is also a chance to start using eco-sustainable techniques and to pursue the circular economy of every country. On the other hand, this paper proposes suggestions to upgrade and improve PMS where it is already in use, considering the progress of road maintenance technology and the concept of sustainable development.

## 2. Aim of the Paper

The definition of a PMS for LRA is of paramount importance, since LRA manage the most road networks and this activity has a huge impact on social and environmental everyday life. LRA significantly differs from highway and airfield administrations and they need a specific PMS based on their real resources and potentialities. Guidelines to be easily applied by LRA have not been internationally shared and defined yet. Indeed, most LRA do not have a running PMS and, in this case, the PMS has to be first identified, selected and adopted.

This paper is intended to enhance the awareness of rational and sustainable administration, even in local contexts, and to describe the procedure as a guideline does, highlighting the need for a coherent method to be implemented or upgraded and adjusted. Therefore, the aim of the paper is to recommend a specific action procedure to apply PMS for LRA using the available or the most suitable technologies.

The recommendations come from many specific experiences, research activities and collaborations with Italian municipalities that manage road networks ranging from 200 to 1000 km, extended by bibliography.

## 3. Objectives and Main Tasks for the Management of a Local Road Network

The main objectives of the PMS for LRA are:

- To determine the planning and design methodology for the management and maintenance of the road network, taking into account the limited human and economic resources. Actions have to be related and coordinated among them in rational order to create a well-defined system to be applied as a routine by the members of the working group;
- To provide different strategies to be compared among them and to allow the most cost-effective and socially advantageous solution to be selected;
- To track and report the motivation of specific investments;
- To monitor and assess the effects and the evolution of a specific project over time. Information recording and analysis help in confirming or changing strategies for the future;
- To improve and standardise road maintenance methods;
- To encourage eco-sustainable solutions, generating development and economic growth by means of circular economy;
- To define roles, competences and responsibilities for technical, administrative and strategic actions;
- To coordinate pavement maintenance, trenching for utilities, accident analysis and every other pavement-related event;
- To increase the technical competence, transparency and reliability of the LRA.

The following issues guide the pursuing of an efficient planning of maintenance works for an LRA:

- Identification of a customized PMS based on rational, objective and sustainable planning approaches;
- Selection of data and implementation of GIS-based tools for the creation and management of the road database;
- Standardisation of the technical procedures to be explained in reference manuals;
- using GIS analysis and decision-making tools to highlight maintenance priorities and to determine the most cost-effective strategy, assessing a set of multi-year investment scenarios;
- Standardisation of repair methods including eco-sustainable materials and techniques;
- Systematic quality control for the road maintenance.

This paper is intended to describe the procedure as a guideline does, highlighting the need for a method and fixing the action flow. The specific tools have to be selected by LRA on the basis of their own potentialities.

## 4. Road Management System for Local Road Administrations

The planning of road maintenance has to be studied considering the user demands and the resources of the LRA. A sequence of functional phases have to be defined for a direct application of the process.

The procedure must be set up as a self-feeding series of interconnected phases, with each activity outcome serving the one after it. Particularly, each functional phase must be specifically linked to actions that must be carried out using customised tools and reference documents. Tools and documents can be developed using high-tech or expeditious and cheap methods, or a combination of them, according to the LRA potentialities and targets.

Figure 1 shows the following functional phases and actions [9]:

- Information: information or data have to be distinguished into static and dynamic data. The static data generally do not change year by year, whereas dynamic data can

change frequently and hence they need a continuous upgrading. The former includes road network classifications (network hierarchy, intersection hierarchy, road function and road type), segmentation of the road network (homogeneous sections, inspection units), road inventory (geometrical data, road elements pavement layers and materials, maintenance and rehabilitation history etc.). The latter involves survey data (traffic volume, pavement condition, emergencies, roughness condition etc.), mobility and accident mapping;

- Maintenance prioritisation: a priority index (PI) must be determined considering the gathered information for every inspection unit, highlighting the areas where a specific action is required and ranking the inspection units on the basis of the PI;
- Type of maintenance: specific repair methods must be standardized and selected depending on distress types, distress severity degree, extension of service life, costs and environmental issues. Non-standard repair methods can be adopted for road sections subjected to high traffic volume or where particular issues have to be considered;
- Multi-years planning: the technical and economical evaluations must be performed using long-term prediction models. Social and environmental impacts have to be assessed as well, comparing different strategies. A strategic index (SI) combining Life Cycle Assessment of projects can be developed and calculated to compare several planning solutions;
- Application: maintenance has to be carried out following the design and specification. Quality control and reporting data have to implement the PMS process.

GIS database and customised-GIS tools facilitate the collection, implementation, analysis and understanding of results by means of thematic mapping [17,18].

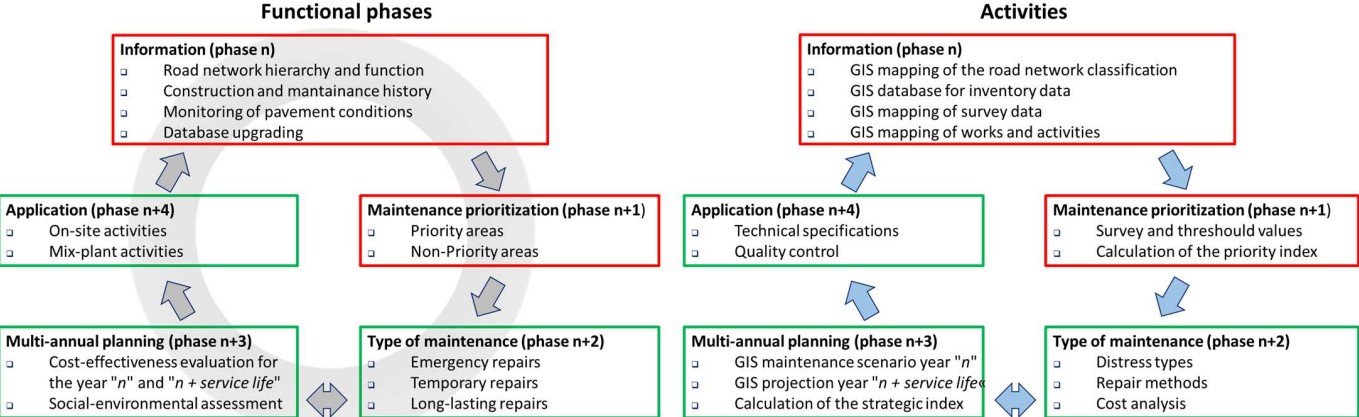

**Figure 1.** Functionality purposes and related actions (red boxes for network level, green boxes for project level).

The management procedure has to be distinguished into two working levels: network and project levels.

At the network level, the management method for the database has to be precisely defined. The survey, analysis and reporting methods have to be described and relative guidelines have to be prepared and available to users. On the entire road network, it is necessary to monitor the condition of the pavement and how pavement distresses change over time. According to technical thresholds, the inspection units with poor conditions or performance must be rated in a list of maintenance priorities.

At the project level, detailed maintenance repair methods of inspection units included in the priority maintenance list must be assessed considering working yards, distress category, severity and size, plant and field organization, environmental impact, technical requirements, service life, budgets and maintenance strategies.

Finally, LRA have to define and adopt rational methods considering their own available human and economic resources. Regardless of the selected methods, which are tightly

related to specific potentialities and resources, the application of systemic and well-defined procedures allows social, technical and environmental benefits to be achieved. Obviously, the more advanced and precise the equipment and tools, the greater the benefits.

## 5. Pavement Management System at Network Level

### 5.1. Digitalization of the Infrastructures and Implementation of GIS-Based Tools for Data Management

An effective PMS requires the creation of a road database in which collecting and storing of different types of data have to take place. Road inventory data can be collected in many formats (hardcopy notes, pictures, videos, spreadsheet tables, automated data acquisition, etc.), which are distributed on a wide territory. In this context, GIS is especially useful to collect and to assign spatial reference to a broad range of heterogenous data to be included in a location referencing system [10]. GIS routines and services for recording, digitalisation and organising of road maintenance data have to be developed, managed and implemented through a dynamic database [9,17].

The road network has to be digitalised and divided into networks and road types, homogeneous sections and inspection units.

Road networks and intersections have to be distinguished and classified according to hierarchical, functional and traffic volume levels. Additionally, the road types have to be classified according to the context (urban or suburban) and the road network level. For example, Figure 2 shows the results of the classification of road network hierarchy and functionalities of roads using GIS mapping for a specific LRA.

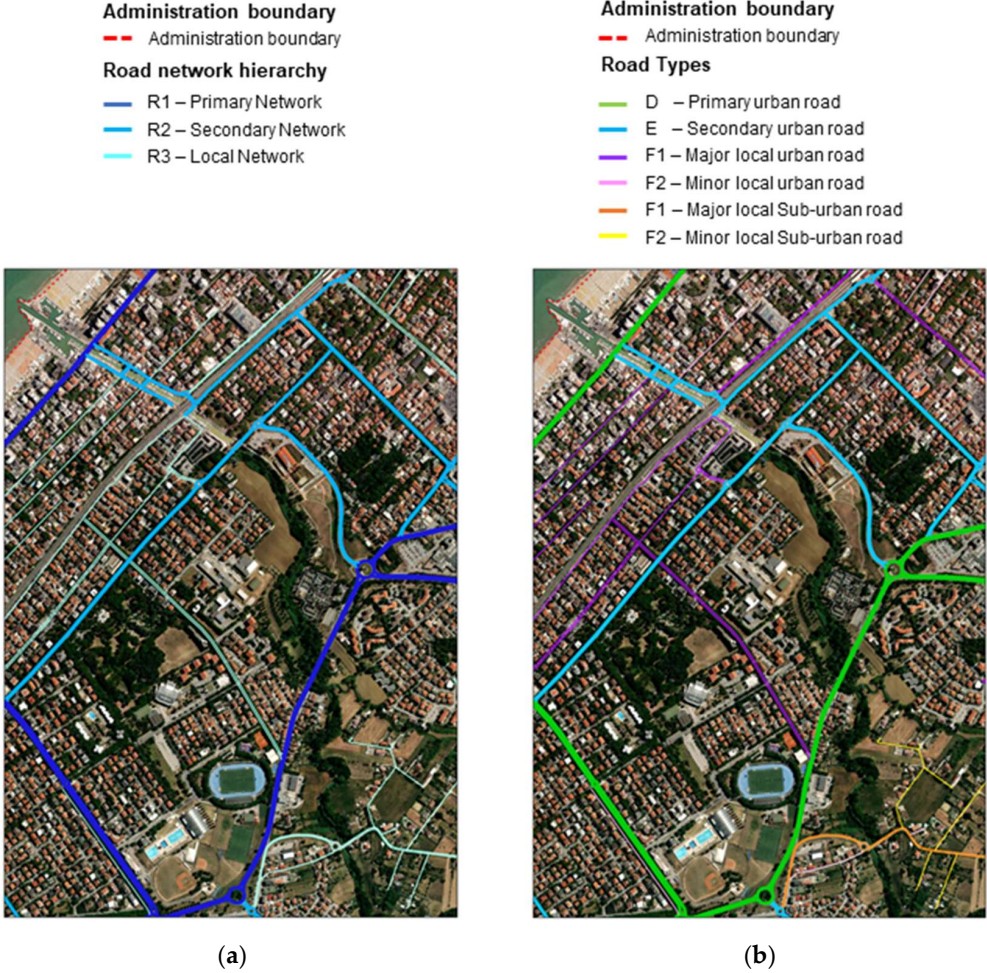

**Figure 2.** Example of GIS mapping of the road network hierarchy (**a**) and functional classification of roads (**b**).

The road network has to be distinguished into easily recognisable and homogeneous sections defined as specific portions of a road with similar and constant properties in terms of: network framework, function, geometric properties, pavement structure (layers and materials), construction date and traffic volume (average daily traffic, percentage of commercial vehicles etc.). Homogeneous sections have to be divided into inspection units for limiting the areas on which pavement information can be gathered. Inspection units such as width as the road lane and can have a maximum length of 0.5 and 1.0 km for urban and suburban roads, respectively.

Road sections and inspection units have to be identified with a unique alphanumerical code to prevent deceptive recognition and analysis in the GIS database.

### 5.2. Road Network Survey and Implementation of GIS-Database

Road survey and monitoring of the pavement conditions have to be performed periodically. Road information upgrading should be performed at least every year for the main road connectors and at least every two years for the secondary and residential road networks.

To rank the roads on the basis of rideability, a rating system has to be defined. Considering the peculiarities of LRA, the visual inspection method can be generally recommended [18,19]. The most common applications refer to specific indexes, such as the pavement condition index (PCI) following the ASTM D 6433-1, [2,7,20], the SN 640 925b, the Belgian Road Research Centre method [21] and the comparison method [22]. Another method merges the use of condition indexes and comparison methods. Several experiences from Italian municipalities suggest classifying the inspection units using a reference scale of pavement conditions based on the calculation of PCI on some specific road sections [16] (Figure 3).

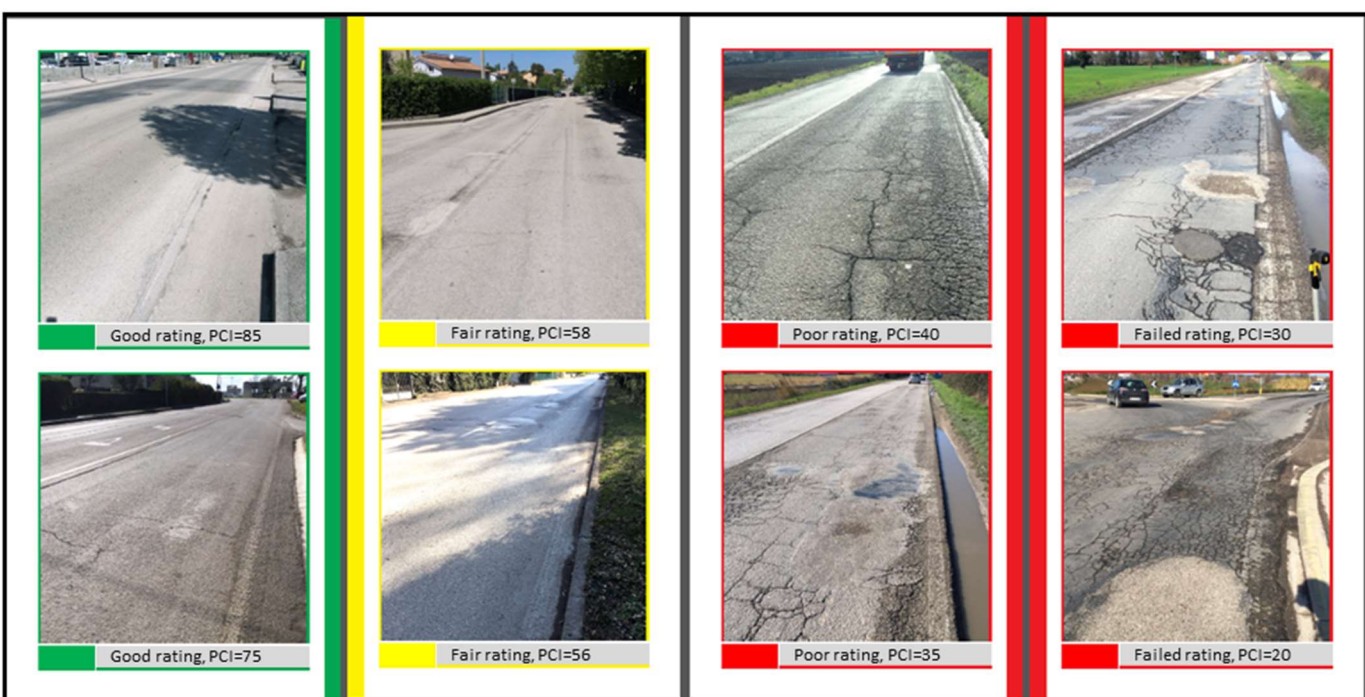

**Figure 3.** Example of a rating system by visual inspection and comparison.

All these methods include a catalogue of distresses that identifies, at least, the type of distress and the measurement methods.

The use of cameras, from recording to the most advanced functionalities, can give a fundamental support to the inventory of road elements, the evaluation of the pavement condition and its evolution over time. When the acquisition system is combined with Global Positioning System (GPS) tracking, video and spatial data can be uploaded in the

GIS database, facilitating the identification of the location of every specific framework. Figure 4 shows an example of combining video framework and GPS location coming from numerous applications for LRA.

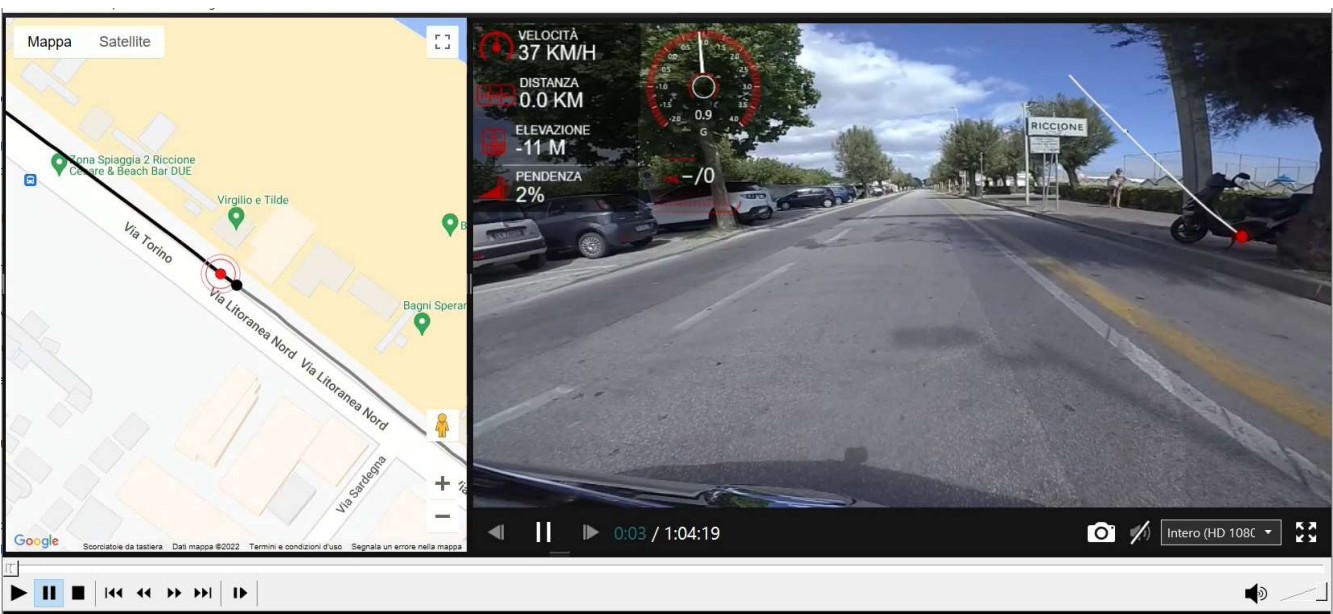

**Figure 4.** Example of a video framework integrated in a GIS database.

Combining visual inspection from spatial-referred video recordings and measurement tools in the GIS database allows the preliminary maintenance work to be designed from the workstation.

Nowadays, easy and cheap methods, supported by verbal descriptors [23], can also be used to measure the International Roughness Index (IRI) and bumps (ASTM E 1926-98) on specific routes using selected applications for smartphones [24,25]. Data can be exported as reports, working sheets or spatial information and, once again, implemented in a GIS database. Spatial-referred video recordings allow emergencies to be identified, as well as road humps to be detected and decoupled from road roughness assessment. In this case, emergencies can include every localized distress and damaged or dangerous road elements (road sign, curbs, hydraulic systems, etc.).

Finally, road inventory and inspection data can be systematically collected with a standard format and shared in the LRA database, using GIS tools. Figures 5 and 6 show an example of how different parameters and results can be represented and highlighted. The GIS dynamic tools allow for combining precise numerical and spatial analysis with a clear and user-friendly display. Users, even without a technical education, can intuitively catch the essence of the plot [26].

A GIS-based service can be created for mobile devices [27] to allow the real-time evaluation and reporting of a road emergency on restricted zones (Figure 7). The mobile service can perform the following in-place operations:

1. Emergency management: photos, description of the distress and recommendation for an emergency repair can be immediately uploaded on a GIS database and an alert containing all the information of the emergency can be directly sent to the road administration office;
2. Management of maintenance works: photos, description and timetable of the works, weather condition and material documentation can be simultaneously uploaded on GIS database;
3. Assessment of road conditions: information, photos, distress details and rating can be uploaded on a GIS database with no need for other operations of transcription.

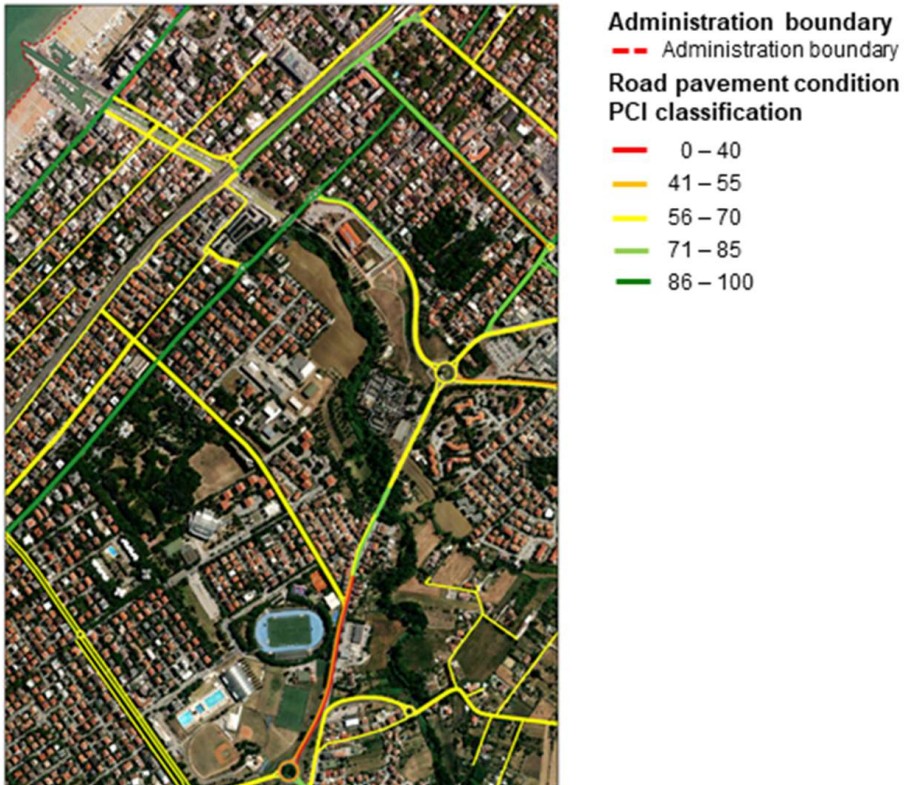

**Figure 5.** Example of GIS mapping of the road pavement condition (PCI classification).

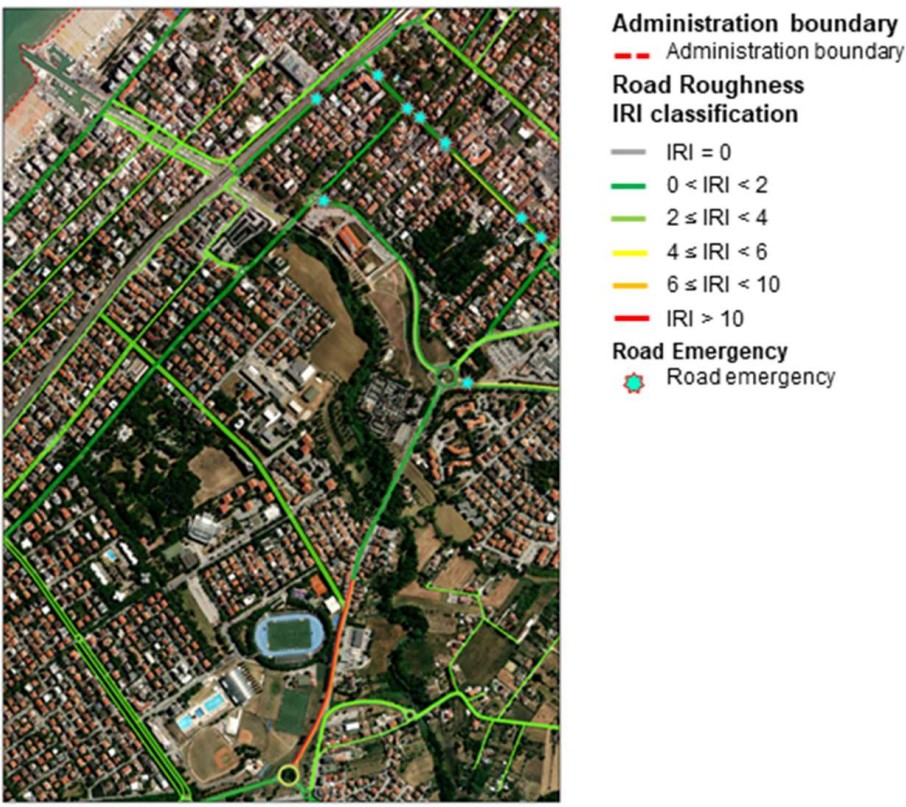

**Figure 6.** Example of GIS mapping of road pavement roughness (IRI classification) and road emergencies.

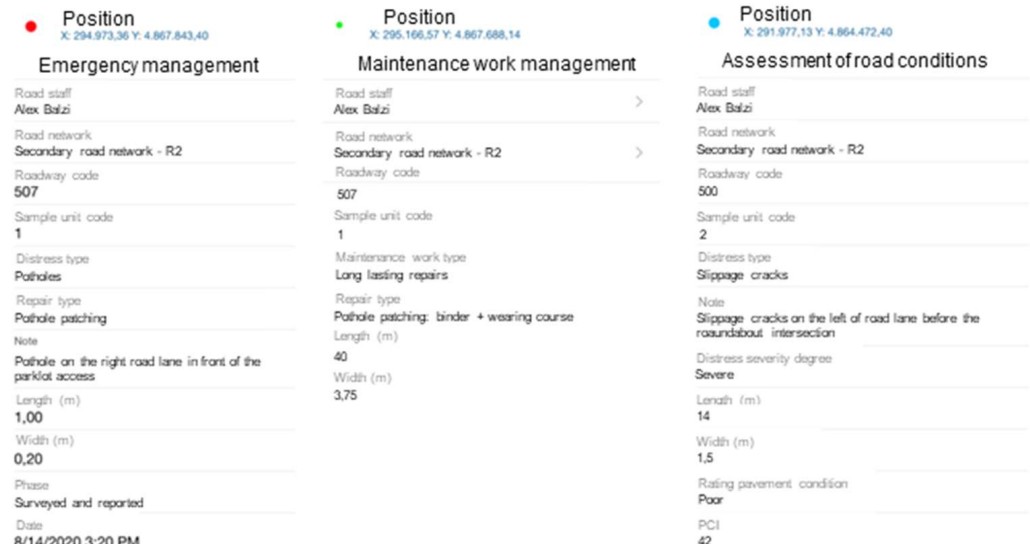

**Figure 7.** Example of GIS-based services on mobile devices for road pavement monitoring and reporting.

### 5.3. Maintenance Prioritisation through GIS Analysis

Planning the maintenance on a wide, heterogeneous and fragmented road network requires the ranking and prioritising working areas. Several parameters have to be established for the description of the condition, functional efficiency and use characteristics. These parameters have to be suitable for the determination of a cost-effective investment scheduling. A priority index (PI) can be defined to support the LRA in the identification of a maintenance priority rating and scheduling of maintenance works. The prioritisation of maintenance should consider the following technical factors [4]:

- Pavement condition to be estimated, measured or calculated (PC);
- Roughness condition to be estimated, measured or calculated (RC);
- Traffic level (TL) based on the average daily traffic values. The percentage of heavy load traffic can also be considered;
- Road function classification and network hierarchy (HF) based on network levels (primary, secondary and local) and road types and functions;
- Strategic function (SF) based on the road relevance: strategical, commercial and tourist routes, places of public interest etc.;
- Historical maintenance (HM) based on average annual costs of previous maintenance works.

Note that the mentioned factors have to be parameterised with appropriate weights using a rating value ranging from 0 to 100. The PI can be set from a combination of the above-mentioned parameters as a number ranging from 100 to 0, where 100 indicates the maximum priority (maintenance needed), while 0 means no repair or suggests preventive maintenance.

The PI has to be calculated if the PC value or RC value, or both, exceed the respective safety thresholds (for instance: PCI < 55, IRI ≥ 6 for primary roads or IRI ≥ 10 for secondary and local roads with vehicles speed lower than 50 km/h).

A linear relationship (Equation (1)) can be used to calculate the PI. LRA have to establish the weight of each parameter considering their own specific needs and strategic addresses.

$$PI = 100 - [x_1 \cdot PC + x_2 \cdot RC + x_3 \cdot TL + x_4 \cdot HF + x_5 \cdot SF + x_6 \cdot HM] \qquad (1)$$

$$\Sigma\, x_i = 1$$
$$0 \leq PC, RC, TL, HF, SF, HM \leq 100$$
PI = 0 when PC and RC respect the threshold for a save rideability

The technical parameters allow us to distinguish roads with similar road conditions (PC and RC) considering use characteristics (TL), role in the road network (HF, SF) and past needs (HM). Based on numerous experiences and collaborations with LRA, the following set of parameters can be suggested: $x_1 = 0.65$; $x_2 = 0.10$; $x_3 = 0.05$; $x_4 = 0.05$; $x_5 = 0.10$; $x_6 = 0.05$) [4,9].

The calculation of PI can be carried out using a GIS-based calculation routine and the available road data on the LRA database. All the priority inspection units can be displayed on the map. Figure 8 shows the results of a specific application for an LRA. All the priority inspection units must be further inspected at the project level to identify the appropriate maintenance method for each area and planning for the road network.

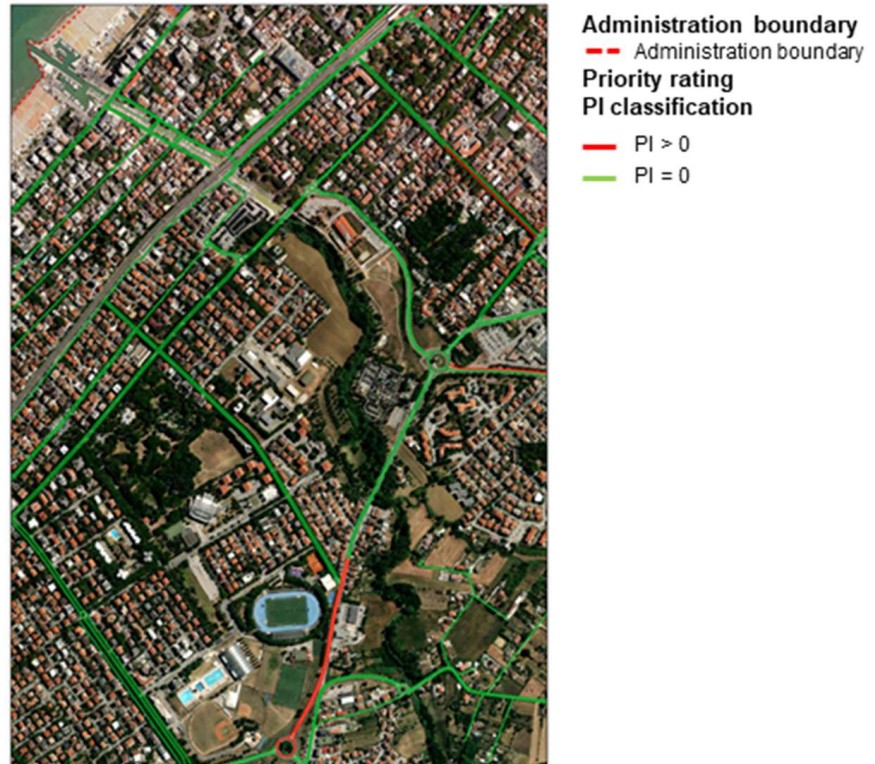

**Figure 8.** Example of GIS mapping of the priority rating (PI classification).

## 6. Recommendation for Using Eco-Sustainable Techniques in the Maintenance of Local Road Networks

### 6.1. General Concepts

Moving from the road network level to project level, LRA should consider eco-sustainable materials, such as recycling techniques and surface treatments as an alternative to traditional techniques to mitigate the environmental footprint of the repair works. Eco-sustainable methods can be quickly introduced as new techniques to rehabilitate road pavements using tailored specifications for LRA [28].

A trial section to be designed by LRA in collaboration with the construction enterprise is recommended when the stakeholders' experience is not solid on eco-sustainable techniques. The trial section should have a minimum size of 350 m² (100 m × 3.50 m and thickness as stated in the design) and a mixture production of at least 90 t to validate the full-scale production of the mixture, to check the production stability and to establish equipment and construction process [29].

LRA can act following coordinated phases to implement new working routines gradually. The following actions are suggested to be implemented in the PMS: specific upgrading lessons for stakeholders, organisation of recycled materials supply, in-lab mix design of the eco-sustainable techniques, building of trial sections at the mix plant, building of

demonstration sites and monitoring of the demonstration site under real traffic volume. If good full-scale monitoring results and social and environmental benefits are obtained, the selected technique can be fully adopted as a maintenance method to be used in the multi-year planning activities [28].

The following chapters show ready-to-use hints and recommendations to introduce recycling techniques and surface treatments in PMS.

### 6.2. Reclaimed Asphalt and Recycled Aggregates to Produce Cold Mixtures

The reclaimed asphalt (RA) and the construction and demolition waste (C&D) should be crushed and screened, producing at least two fractions to reduce the heterogeneity of particle gradation and physical properties. RA and C&D fractions should be stockpiled and covered using open air sheds to avoid direct sun irradiation and wetting by precipitation. Each fraction of RA should be identified by average and coefficient of variation values of the characteristics required in the procedure used for mineral aggregates including the gradation of RA (wet sieving) and inner aggregates, shape index, flakiness index and bitumen content. Whilst each fraction of C&D should be characterized by average and coefficient of variation values following the procedure used for mineral aggregates and declaring the constituents of recycled aggregates. Particularly, clay masonry units, calcium silicate masonry units, aerated non-floating concrete, glass, clay and soil, metals, wood, plastic and rubber, gypsum plaster and floating particles should not exceed 10% by volume of the total mixture, and plasticity index should be less than 6. Statistics should take into consideration at least five samples when the total amount of RA or C&D to be treated in the project is less than 2500 t, or 1 sample every 500 t when the production employs a higher RA or C&D quantity. The RA and C&D contents should be declared for every mixture. Recycled blends should be checked daily, or every 5000 m$^2$ of paving, to control the production stability [29].

### 6.3. Hot Recycled Asphalt Concrete

For hot recycled asphalt concrete (HRAC), an appropriate and reliable organization and quality control of RA and products should be developed to apply no restriction on the amount of RA that can be recycled. This strategy leads to an accurate RA management, production chain innovation and hot recycling process optimisation without compromising mixture performance. The specifications for the constituent materials and performance should be the same as for conventional asphalt concrete. However, tight requirements for RA, rejuvenators and products should be established during the quality control phase [30,31]. Since base courses can be constructed using cold procedures, HRAC can be utilized for wearing and binder courses [28].

### 6.4. Cement-Treated Recycled Mixture

The cement-treated recycled mixture (CTRM) can include up to 100% of RA and C&D using mineral aggregates for the improvement of the gradation only when needed [32]. CTRM can be manufactured in situ, also involving marginal soils, or in mobile or stationary mixing facilities. Typically, CTRM is applied for foundation and base courses or as filling material in utility trenches. The granular blend (not including cement) should ensure a CBR index higher than 50 and no vertical swelling after 96 h of immersion in water. A specific mix design should be adopted to reach indirect tensile strength higher than 0.30 N/mm$^2$ and unconfined compressive strength higher than 3.00 N/mm$^2$ after 7 days curing time at 25 °C [32].

### 6.5. Cold Recycled Asphalt Mixture

The cold recycled asphalt mixture (CRAM) can include up to 100% RA. Mineral aggregates and filler can be added to correct the gradation when required. CRAM can be produced in place and in mobile or stationary mixing facilities [29]. A combination of a

bituminous binder (in form of foam or emulsion) and cement can be used to design a wide range of mixtures to be applied as binder and base layers [33].

A specific mix design should be adopted to achieve indirect tensile strength higher than 0.40 N/mm$^2$, indirect tensile strength ratio higher than 80% and indirect tensile stiffness modulus higher than 3000 MPa at 20 °C after 3 days curing time at 40 °C [28,34].

### 6.6. Surface Dressing

Even if surface dressing can be manufactured by only one layer of binder and one layer of chippings, a long-lasting surface dressing should be designed considering three layers of binder and chippings to appropriately seal the pavement, to improve the skid resistance and to harmonize the road in a rural context. The dosage of bituminous emulsion should be about 3.0 kg/m$^2$ for the first layer and 1.5 kg/m$^2$ for the second and third layers. An aggregate size of 8/12 mm and a dosage of 10 L/m$^2$ should be used to saturate the first and second layer, whereas an aggregate size of 4/8 mm and a dosage of 7 L/m$^2$ should be used on the surface [28].

### 6.7. Microsurfacing

Microsurfacing can be applied as one or two layers over old or new pavements [35]. When two layers must be laid down, a 0/4 mix should be applied as the sealing and levelling layer (first layer) and a 0/8 mix should be applied on top (second layer) to provide a surface with high skid resistance. The microsurfacing recipe should be fixed following a specialised mix design to obtain specific properties such as: wearing after 1 h of immersion lower than 500 g/m$^2$, cohesion after 30 and 60 min higher than 1.2 and 2.0 N·m, respectively, consistency between 25 and 35 mm [28].

### 7. Conclusions

This paper describes concepts, practices and tools that can be easily developed by a Local Road Administration (LRA) for a tailored Pavement Management System (PMS).

The process must be organised as a series of interconnected phases, where each functional phase is linked to a specific set of tasks. Every task must be carried out using a certain set of tools and reference documentations.

The standardisation and rationalisation of the recording, digitisation and management of road maintenance data in a shared database can be made using GIS-based solutions. The graphical visualisation of the information on road network hierarchy, road network functions, road type, historical data, survey outcomes in thematic maps simplify the assessing and the interpretation of the results. Priority set, repair methods and cost-effective analysis can be performed using GIS-based tools. The quality control and reporting data of the planned maintenance works can be collected, ensuring the systematic updating of the GIS database.

Furthermore, LRA can provide a significant contribution to both economic and environmental challenges, proposing a gradual shifting from conventional to eco-sustainable techniques though a tailored PMS. LRA can indeed promote the changing of the way of production and consumption of goods and resources. Through PMS, LRA can impose an efficient management of natural resources and disposal of waste to encourage recycling, reducing waste and developing strategies to limit the increase in global mean temperature.

The comprehensive activity opens new prospects for the management and maintenance of the road network. Even though huge advantages can be only attained gradually, the use of tailored PMS leads to improvement of quality standards, determination of good practices, help and validation of the maintenance strategies.

**Author Contributions:** Conceptualization, A.G.; methodology, A.G.; software, A.B.; validation, A.B.; formal analysis, A.G.; investigation, A.B.; resources, A.G.; data curation, A.B.; writing—original draft preparation, A.G.; writing—review and editing, A.B.; visualization, A.B.; supervision, A.G.; project administration, A.G.; funding acquisition, A.G. All authors have read and agreed to the published version of the manuscript.

**Funding:** This work was supported by the University of the Republic of San Marino, Research project UniRSM 2021 "Riutilizzo dei materiali da demolizione e delle terre da scavo per l'economia circolare delle opere di costruzione e manutenzione delle infrastrutture viarie: sviluppo e proposte di applicabilità nella Repubblica di San Marino".

**Data Availability Statement:** The data used to support the findings of this study are contained within the article. More details are available on request from the corresponding author.

**Acknowledgments:** Authors express their gratitude to the University of the Republic of San Marino for the financial support. In addition, the authors wish to thank the Azienda Autonoma di Stato per i Lavori Pubblici (Repubblica di San Marino), Ufficio Pianificazione Territoriale, (Repubblica di San Marino), Comune di Fano (Italy), GEAT (Riccione, Italy), Comune di Riccione (Italy), Comune di Ravenna (Italy) for the collaboration, interactive exchange of knowledge and opinions.

**Conflicts of Interest:** The authors declare no conflict of interest.

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
