# Peer review of "Methodologic Recommendations to Implement Pavement Management Systems and Eco-Sustainable Solutions for Local Road Administrations"

_infrastructures, doi:10.3390/infrastructures8020025_

Round 1
Reviewer 1 Report (Previous Reviewer 2)
The paper was improved in many areas. Nevertheless, the reviewer finds it difficult to understand why this paper is considered as a research article. For instance, the methodology presented in section 4.3 should have been applied and validated followed by the presentation of the results (even as a case study). Please reconsider.
In addition, section 5 provides indeed valuable comments, but again did they come as a result of the authors' own research or as a result of literature review. Please reconsider.
Author Response
Please see the attachment.

Reviewer 2 Report (Previous Reviewer 1)
The manuscript is much improved now. However, some crucial references are missing. The manuscript can be accepted subject to the following minor revision:
Update the following relevant references: [Transp. Res. Rec. J. Transp. Res. Board 2012, 2299, 41–47; ISPRS Int. J. Geo-Inf. 2017, 6, 24; Infrastructures 7 (9), 114, 2022].
Author Response
Please see the attachment.

Reviewer 3 Report (New Reviewer)
Although pavement management and management system has been widely used in road infrastructure management, with the progress of road maintenance technology and the proposal and improvement of the concept of sustainable development, the content, method and management system of pavement management need to be constantly upgraded and improved. In this sense, this article is meaningful. However, this paper only introduces the concept and working principle of the pavement management system, as well as relevant maintenance treatments,and lacks research and innovation. It is suggested that the author submit it after modification.Author Response
Please see the attachment.

Round 2
Reviewer 1 Report (Previous Reviewer 2)
The content of the paper is now OK with constuctive new text. I have again concerns about the type of this submission, which is considered as ARTICLE (line 1 in the paper). Please see these types: https://www.mdpi.com/about/article_types and probably consult the editorial office about which is the correct submission type for your work (e.g., is it an "opinion" article?).
Author Response
R: The content of the paper is now OK with constuctive new text.
A: Authors thank this reviewer. The comments allow the authors to explain much better topics and approach and to improve many details in the paper.
R: I have again concerns about the type of this submission, which is considered as ARTICLE (line 1 in the paper). Please see these types: https://www.mdpi.com/about/article_types and probably consult the editorial office about which is the correct submission type for your work (e.g., is it an "opinion" article?).
A: Unfortunately authors did not realise the list of the type of submission before. As the editorial department suggests, authors agree in considering the paper as a technical note.
Reviewer 3 Report (New Reviewer)
The author has revised the article and submitted the modification instructions. From the perspective of promoting and popularizing PMS, the revised content of this article is meaningful and agreed to be published, although the PI calculation of PMS does not reflect the eco-sustainable solutions and its technical advantages.
Author Response
R: The author has revised the article and submitted the modification instructions. From the perspective of promoting and popularizing PMS, the revised content of this article is meaningful and agreed to be published,
A: Authors thank this reviewer. The comments helped the improvement of the paper significantly. Topic and discussion are remarkable clearer.
R: although the PI calculation of PMS does not reflect the eco-sustainable solutions and its technical advantages.
A: Authors agree to the reviewer. PI does not reflect the eco-sustainable solutions and has not to reflect them. The PI (priority index) values highlight were the design has to be developed. After that, a detailed analysis and the design have to evaluate eco-sustainable solutions. The detail of this second part will be published in the next future by the authors.
The paper reports (pag. 4): “multi-years planning: the technical and economical evaluations must be performed using long-term prediction models. Social and environmental impacts have to be assessed as well, comparing different strategies. A strategic index (SI) combining Life Cycle Assessment of projects can be developed and calculated to compare several planning solutions”
This manuscript is a resubmission of an earlier submission. The following is a list of the peer review reports and author responses from that submission.
Round 1
Reviewer 1 Report
This paper shows the approach, skills and processes for the PMS gathered over time from several collaborations with LRA. Moreover, this article outlines the strategy and requirements for environmentally friendly road materials, including cold and hot recycled asphalt concrete, cement-treated recycled materials, and cold surface treatments, which can be progressively adopted by each LRA to lessen the negative effects of maintenance activities on the environment.
The results are interesting and have some novelty. However, a revision is needed.
1. What is the motivation and justification for considering the weights as described in equation (1)?
2. What are the limitations of this method described in this paper?
3. How is the existing knowledge impacted because of this current study?
4. Elaborate more clearly in the abstract what the original contributions of this manuscript are.
5. The overall presentation needs improvement.
Reviewer 2 Report
To the reviewer’s opinion, the theme is worthy of investigation and suits the journal's aim. However, it is not clear how this paper does contribute to the existing state-of the art. At first, it seems that the paper gathers existing practices in local road management, whereas in the second part, already-known sustainable approaches are presented. The authors should reconsider their aim and compile individual sections in a way that ensures cohesion and consistency. A probable reconsideration of the submission type might be also needed (is this really a research article and with what research findings?).
Reviewer 3 Report
This manuscript mainly describes the functions and methods of the payment management system and several green maintenance technologies. It can be considered an introduction to a system or method, rather than a normal scientific paper. There is lack of key problems, methods, algorithms, data and analysis processes. Therefore it cannot be accepted as a scientific paper.
It is suggested to give a clear aim of the problem to be solved, and propose specific methods and algorithms, and carry out actual analysis rather than description in words.
Specifically, 1. In Figure 5 and Figure 6, it is necessary to explain in detail how different classifications are implemented on the routines, whether they can be dynamically displayed or what algorithms are used. 2.In Eq. (1) How to obtain the weight of each parameter? 3. What is the relationship between recommendation for using eco sustainable technologies and the payment management system? How do they support each other? Specific data and correlation verification are required.